# Immortalization Reversibility in the Context of Cell Therapy Biosafety

**DOI:** 10.3390/ijms24097738

**Published:** 2023-04-23

**Authors:** Oksana I. Sutyagina, Arkadii K. Beilin, Ekaterina A. Vorotelyak, Andrey V. Vasiliev

**Affiliations:** N.K. Koltzov Institute of Developmental Biology of Russian Academy of Sciences, Laboratory of Cell Biology, Vavilov Str. 26, 119334 Moscow, Russia

**Keywords:** reversible immortalization, hTERT, SV40T, c-MycER^TAM^, Tet-On/Tet-Off system, site-specific-recombination, CRISPR/Cas9, suicide genes

## Abstract

Immortalization (genetically induced prevention of replicative senescence) is a promising approach to obtain cellular material for cell therapy or for bio-artificial organs aimed at overcoming the problem of donor material shortage. Immortalization is reversed before cells are used in vivo to allow cell differentiation into the mature phenotype and avoid tumorigenic effects of unlimited cell proliferation. However, there is no certainty that the process of de-immortalization is 100% effective and that it does not cause unwanted changes in the cell. In this review, we discuss various approaches to reversible immortalization, emphasizing their advantages and disadvantages in terms of biosafety. We describe the most promising approaches in improving the biosafety of reversibly immortalized cells: CRISPR/Cas9-mediated immortogene insertion, tamoxifen-mediated self-recombination, tools for selection of successfully immortalized cells, using a decellularized extracellular matrix, and ensuring post-transplant safety with the use of suicide genes. The last process may be used as an add-on for previously existing reversible immortalized cell lines.

## 1. Introduction

Currently, cell therapy and tissue engineering are widely used to overcome the challenges of donor material shortage, resolve incompatibilities between donor and recipient tissues, provide a personalized therapeutic approach, and incorporate genetically modified cells with specific traits into therapy. Within the framework of these modern therapeutic strategies, cells are transplanted, alone or as a part of a tissue-engineering construct (tissue-engineered skin equivalent, for example [1]), or are used as a main component in bio-artificial organs (such as bio-artificial liver [2], bio-artificial kidney [3], and lung-assist device [4]). However, cell therapy and tissue engineering require a considerable number of cells (up to 7 × 10^7^–6 × 10^10^ cells per patient [5,6]), and primary cells derived from non-cancerous tissues have limited proliferation capacity due to replicative senescence.

Immortalized cells are a promising source of cells for therapeutic application. During immortalization, cells undergo genetic remodeling, which helps to avoid replicative senescence. Therefore, immortalized cells (including primary) acquire an unlimited ability to divide [7]. After the required cell mass has been obtained, immortalization is reversed to “switch down” the immortalized state: cells stop unlimited proliferation and return to a mature phenotype, which was temporarily lost during immortalization. This is necessary for the restoration of their functions during in vivo use [8]. For example, hepatocytes and pancreatic beta cells regain the ability to produce liver enzymes and insulin, respectively, after the reversal of immortalization [2,8]. Thus, immortalization makes it possible to obtain an unlimited number of cells with the required phenotype.

However, this approach comes with certain risks. As already pointed out, the clinical use of cells requires the immortalization process to be “switched down” before they can be safely used in a clinical setting. This is necessary not only because of the need to return to a mature cell phenotype but also because of the danger that uncontrollably dividing cells would be present in the human body. If transplanted (or if they escape into the patient’s bloodstream from bio-artificial devices coming in contact with blood), such cells may participate in tumor development or undergo a malignant transformation in vivo due to the use of oncogenes or oncoproteins as immortalizing agents [4]. Immortalization reversal solves this problem, but is it certain that immortalization reversal is successful in 100% of cells? In addition, an immortalization–de-immortalization process may potentially be accompanied by unwanted side effects in the cell. Even a single cell predisposed to or capable of uncontrolled growth and malignant transformation can start the oncogenic process or neoplasm formation, which makes the problem serious. However, as will be shown below, this is not a reason to avoid using the method.

As an alternative to immortalized cells, stem cells—and especially induced pluripotent stem cells (iPSC)—can be used because of their capacity to proliferate indefinitely and generate a wide range of specific differentiated derivatives [9,10,11]. However, adult stem cells (embryonic stem cells are not used due to ethical restrictions) are available in limited quantity, have a limited life span in culture [12,13], and their long-term proliferation is associated with the decline of the differentiation potential [12,13,14]. Mass production of iPSCs, in turn, is associated with significant difficulties [6]. Due to the inherent features of iPSCs, they cannot be cultivated in standard bioreactors (thus, the development of specific equipment is required) and require an original culture medium with expensive components, of which thousands of liters may be required for mass production. While systems for mass iPSC production are currently being developed [6,15], iPSCs for therapeutic applications are still “hand-made” in laboratories (which cannot provide the number of cells required for the widespread use of cell therapy) [6,15], and as of 2021, the cost of a single therapeutic dose of iPSC-derived cells was around US $1 million [15]. Immortalized cells, compared to iPSCs, are less delicate, and their mass production does not require the development of specific equipment and a highly specific culture medium, making the process less complicated and less expensive.

Conditions used for the differentiation of iPSCs are far from being selective and include a combination of growth factors for each cell type. The slightest inconsistency in cultivation conditions negatively impacts iPSC differentiation and the quality of the final cell product [6,15]. In contrast, a wide range of cell types, including those that are highly differentiated, can be subjected to immortalization. Relevant developments already exist and show promising therapeutic effects; some examples are presented in Table 1.

According to the data presented in the table above, immortalization has been successfully applied not only to primary somatic cells but also to blast cells (Table 1, No. 6, 9) as well as progenitor cells (Table 1, No. 7–8) and multipotent stem cells (Table 1, No. 1–2). Stem cell immortalization provides a solution to the problem of differentiation-potential decline described above: immortalized stem cells in most cases retain their differentiation ability during serial passaging [12,13,14].

Immortalization of genetically modified cell lines requires special mention. This process allows us to produce a significant number of genetically engineered cells with specific properties, such as astrocytes that are capable of synthesizing and releasing GABA in the substantia nigra under the control of a tetracycline-sensitive promoter, which can be used for the treatment of Parkinson’s disease (Table 1, No. 5), modified muscle progenitors carrying human artificial chromosomes with dystrophin locus (DYS-HAC), used for Duchenne muscular dystrophy therapy (Table 1, No. 8), insulin-producing hepatocytes capable of glucose-dependent insulin release (Table 1, No. 14), or human renal proximal tubule epithelial cells with OAT1 overexpression (OAT1 protein plays a central role in the renal organic anion transporter), which may be more effective as a cell component in a bio-artificial kidney (Table 1, No. 20). In this case, immortalization makes it possible to multiply genetically modified cells in the required quantities, which cannot be achieved through the use of iPSCs.

Thus, immortalization is a convenient and profitable method with great potential and a wide range of applications; therefore, the presence of risks associated with the method does not constitute grounds for refusing to use it (in favor of using iPSCs, for example) but must be considered a reason for modifying it. It should also be noted that, as can be seen in the table, most of the studies related to the use of reversibly immortalized cells are still in progress or at the stage of preclinical trials; thus, the question of the modification method is timely.

The aim of this review is to systematize various approaches to reversible immortalization, analyze possible side effects, highlight associated risks, and suggest possible tools for risk reduction. We hope that this review will facilitate the choice of the safest immortalization–de-immortalization strategy for those who are just starting to work with immortalization and encourage the use of add-on-tools to improve the safety of the process for those who already have stable lines of immortalized cells. In order to identify the potential risks associated with immortalization–de-immortalization, it is necessary to understand the mechanisms of these processes. Relevant information is briefly presented in the next two sections.

## 2. Immortalization of Cells

As mentioned above, the main aim of immortalization is to overcome cell replicative senescence. A critical component of cell replicative senescence is telomere dynamics [7]. Telomere-associated senescence is caused by the shortening of telomeres that occurs with each human somatic cell division, and most human cells do not maintain sufficient telomerase activity for effective maintenance of telomere length [7,58]. Eventually, telomere shortening leads to cell cycle arrest. Overriding cell cycle arrest and constant telomere elongation are two approaches used during immortalization [7,59]. The insertion of viral oncogenes (*SV40T* [60], human papillomavirus *E6* and *E7* oncogenes [60,61]), *Myc* oncogene family members [62], *BMI1* (a *c-Myc* cooperating oncogene) [63,64]), and human telomerase reverse transcriptase gene (*hTERT*) [7,58] are usually used for the expression of the corresponding proteins that are immortalization actors. The action of these agents is different.

Viral oncoproteins—simian virus 40 large T antigen (SV40T) and human papillomavirus E6 and E7 oncoproteins (HPV16 E6/E7)—induce immortalization by p14^ARF^/p53 and p16^INK4A^/pRb pathway inhibition [65], which are two main pathways responsible for cell cycle arrest in replicative senescence [66]. SV40T [67,68] binds to the pRb-E2F complex, leading to the dissociation of transcription factor E2F from the SV40T-pRb complex. In the absence of binding with pRb, E2F1 mediates transcription of E2F1-target genes that causes G1/S transition and S-phase entry [1,68]. SV40T also binds to p53, thereby suppressing the p53 pathway [67,68]. Similarly, HPV16 E6/E7 (currently not widely used) inactivate the p53 and pRb proteins, respectively [69]. Thus, the use of oncoproteins allows us to overcome the problem of cell cycle arrest but not of telomere shortening.

Myc oncoproteins and BMI1 provide both telomere length maintenance and cell cycle activation. Myc family oncoproteins (c-Myc [20], v-Myc [70], L-Myc [71], N-Myc [72]) stabilize telomere length via the upregulation of hTERT expression and prevent the accumulation of cell cycle inhibitory proteins, such as p16^INK4a^, which normally prevent S-phase progression by binding CDK4/6 and inhibiting cyclin D–CDK4/6 complex formation [73]. BMI1 prolongs the replicative lifespan by suppressing the p16Ink4a-dependent senescence pathway and increasing hTERT expression [64,74].

The third type of immortalizing agent, hTERT, is a catalytic subunit of human telomerase that synthesizes new telomeric DNA from the RNA template [75]. TERT was shown to be the major limiting agent for telomerase activity [76], so hTERT cDNA transfer provides constitutive telomere maintenance [77] without cell cycle overactivation.

SV40T and E6/E7 HPV16 or hTERT expression alone may be insufficient for immortalization. Although *hTERT* insertion alone is enough to immortalize a range of primary cells [77,78], it is insufficient for the immortalization of others, such as keratinocytes, which require additional activation of cyclin-dependent kinases to pass through checkpoints in the mitotic cycle [79]. Using only *SV40T* [59] or *E6/E7 HPV* [58] insertion for immortalization (without *hTERT* gene), researchers were able to suppress the p53/p21 and p16/Rb pathways without telomere extension, which leads only to the so-called “pre-immortal” state (prolongation of cell proliferative ability) but not to the real “immortal” state (unlimited proliferation ability) [59,77]. Suppression of p53/p21 and p16/Rb allows us to overcome cellular growth arrest (known as mortality stage 1, M1) [59,77], but the telomeres continue to shorten [7], which eventually leads to so-called mortality stage 2, M2, characterized by proliferation arrest and extensive cell death [59,77]. Due to this issue, a combination of immortalizing genes with different effects is usually used.

Thus, there are three different mechanisms of immortalizing agents: providing cell cycle arrest without maintaining telomere length, combining the maintenance of telomere length with cell cycle activation, and, finally, maintaining telomere length without direct cell cycle control. As shown below, that how exactly does the immortalizing agent works is important in the context of the problem considered.

## 3. Reversible Immortalization: Approaches and Tools

Currently, many different approaches are used for reversing immortalization, including tools developed in the late 1980s as well as modern ones. However, the data presented in the literature are scattered and need to be systematized.

The approaches to reverse immortalization can be divided into three groups. The basic principle of the first group is that the immortalization state is condition-dependent, and changing conditions causes de-immortalization. The basic principle of the second group is the excision of an immortalizing genetic element by site-specific recombination. The third group involves the use of small interfering RNAs (siRNAs) for gene silencing (Table 2).

### 3.1. Conditional Immortalization

Conditionally immortalized cells remain in an immortalized state under certain conditions and lose that state when the conditions change.

#### 3.1.1. Temperature-Dependent Reversible Immortalization

The first example of conditional immortalization involves the use of a temperature-sensitive mutant SV40T form (tsA58 SV40T) [49,50,51,80,81,82,91,92,93,94,95,96] (Table 2, No. 1). The tsA58 SV40T is active and maintains cell division at 33 °C (so-called permissive conditions) and switches off the “immortalized program” at a higher body-specific temperature (37 °C) when culture temperature is changed in vitro or cells are transplanted in vivo (non-permissive conditions) [91,92,93]. Immortalized cell lines are routinely generated by transduction of the cells of interest with a *tsA58-SV40T*-carring retroviral vector. Moreover, to avoid the need for gene insertion in vitro to acquire *tsA58 SV40T* reversibly immortalized cell lines, a special strain of transgenic mice H-2Kb-tsA58 was created known as Immortomice^TM^ [93]. These mice carry a transgene that expresses the *tsA58 SV40T* under the control of the interferon-inducible murine H-2Kb promoter, which is present in a homozygous form to be able to transmit a functional copy of the transgene [93,94]. Immortalized cells can be isolated directly from an H-2Kb-tsA58 mouse simply by dissociation of the tissue of interest. First, they are cultured at 33 °C to obtain the required number of cells and then at 37 °C to provide the differentiation into various phenotypes as described previously [36,80,81,82,93,95,96]. This tool may be very useful for the generation of specific cell cultures (e.g., Kupfer cells [81], cardiac endothelial cells [82], etc.) and their potential clinical applications. In general, it has been shown that this approach is suitable for the reversible immortalization of a wide range of cell types, including renal epithelial cells [49,50,51], myoblasts [36], alveolar type II cells [80], hippocampal cells [23,24,25], auditory neuroblasts [32,33], retinal progenitor cells [34], and renal epithelial cells [29,30,31].

#### 3.1.2. Tamoxifen-Dependent Reversible Immortalization

Both human c-Myc [97] and viral v-Myc [98] oncoproteins may be made functionally dependent on the synthetic estrogen receptor ligand, 4-hydroxytamoxifen (4-OHT), via fusion with a mutant murine estrogen receptor [16,17,18,19,83,84,85,97,98,99,100,101]. While v-MycER construction is very rarely used, c-MycER is a widely used tool (Table 2, No. 2). c-MycER^TAM^ is a recombinant protein that represents the fusion of a human c-Myc and a modified mouse estradiol receptor (ER) [97]. In the absence of 4-OHT, c-MycER^TAM^ is present in its inactive (monomer) form in the cell cytosol. When added to the culture medium, 4-OHT specifically binds to the ER, causing c-MycER^TAM^ dimerization and translocation to the cell nucleus, where c-Myc is active as a transcription factor, maintaining cell proliferation [97,98]. The withdrawal of 4-OHT leads to c-MycER^TAM^ inactivation [97,100,101]. It is important to note that due to structural modifications, the mouse estradiol receptor is unable to bind its natural ligand, 17β-estradiol, so there is no risk of accidental Myc activation in vivo [100,101]. This approach is suitable for the reversible immortalization of various cell types, including neural stem cells [16,17,18,19], olfactory mucosa cells [83], embryonic stem cells [84], and hepatocytes [85].

#### 3.1.3. Tetracycline-Dependent Reversible Immortalization

Tetracycline-controlled systems [102] are widely used to regulate gene transcription in eukaryotic cells. These systems consist of two components: an activation domain of a transcription factor fused to TetR, forming a transcriptional activator, and a TET promoter fused to the tetO sequence, regulating the target gene [41,102,103,104]. There are two types of tetracycline-regulated systems: tetracycline-repressible (so-called Tet-Off) [102] and tetracycline-inducible (so-called Tet-On) [105]. In the Tet-Off system, the transactivator binds to the tetO component, inducing gene expression in the absence of tetracycline [102]. Therefore, Tet-Off allows the silencing of gene expression by the addition of tetracycline or its derivatives. In contrast, in the Tet-On system, the reverse transactivator binds to the tetO sequence in the presence of tetracycline, allowing activation of gene expression by tetracycline [105]. Tetracycline-controlled systems are used to regulate immortalizing gene (*v-Myc* [89], *SV40T* [86,87], and *hTERT* [88]) expression. Both Tet-Off [101] and Tet-On [86,88] systems are used for reversible immortalization (Table 2, No. 3); however, Tet-On seems to be more suitable for this purpose, as it does not require constant treatment with tetracycline or its derivatives to stop cell division. The Tet-Off system can be used in cell models but not in therapy. This approach is suitable for the reversible immortalization of different cell types, including neuronal progenitor cells [89], granulosa cells [86], conjunctival epithelial cells [87], bone mesenchymal stromal cells [88], and pancreatic beta cells [41].

### 3.2. Site-Specific Recombination

The second group of methods provides immortalization reversibility via excision of the immortalizing gene. These methods involve retrovirus-mediated, transposon-mediated, or CRISPR/Cas9 HDR-mediated transfer of the immortalization gene that is subsequently excised by site-specific Cre-Lox or FLT-FRP recombination. This is currently the most commonly used and actively developing strategy (Table 2 No. 4, 5).

Immortalizing genes—*SV40T* [40], *hTERT* [42], or their combination (*SV40T* + *hTERT* [38,39,106])—are flanked by two homodromous LoxP sites [107,108] or two flippase recognition target (FRT) sites [109,110,111,112,113] and are delivered by a retroviral vector [107,108,110,111,114,115], CRISPR/Cas9 system [54], or by the PiggyBac transposon system (transposon-mediated vector pMPH86) [55,112,113], which is one of the two most widely used transposon systems (PiggyBac, Sleeping Beauty) [56,116,117] and exhibits higher transposition activity than Sleeping Beauty in cultured human cells [117].

To knock out immortalizing genes before in vivo application, cells are transduced with Ad-Cre [56] or Ad-FLP [118,119] adenoviral vectors or relative plasmids so that immortalized genes are specifically removed by Cre or FLT recombinase, respectively.

### 3.3. Small Interfering RNA-Mediated Immortalizing Gene Silence

Small interfering (siRNA)-mediated immortalizing gene silencing (Table 1 No.21) is a novel trend. Within the RNA interference mechanism, siRNA can knock down the expression of target genes in a sequence-specific way by mediating targeted mRNA degradation [120]. *HPV16 E6/E7-* [121], *c-Myc*- [122], *SV40T*- [123,124], and *hTERT* [125]-specific siRNAs are commonly used as therapeutic (mainly anti-cancer) tools but can also be utilized to reverse immortalization. siRNA-*SV40T* [52] or a combination of siRNA-*SV40T* and siRNA-*hTERT* (if cells are immortalized with both genes) [126] are currently used as de-immortalization agents. *E6/E7* siRNA is used to study the role of E6 and E7 in telomerase regulation [127] but can also be used for de-immortalization. Recent studies have shown that siRNA-mediated immortalizing gene silencing is a suitable approach for immortalization reversal. This approach (as a de-immortalization strategy) has been successfully applied to mouse keratinocytes [52] and human pancreatic beta cells [126].

## 4. Reversible Immortalization: Possible Risks and Ways to Avoid Them

Considering the issue of biosafety of the reversibly immortalized cells, two basic risks should be distinguished. The first risk involves the possibility that in some cells, de-immortalization does not occur, and cells that retain the immortalized status (so are able to divide indefinitely) are transplanted into the patient’s body. The second risk is that manipulations in the process of immortalization–de-immortalization lead to negative changes in the cell—foremost, to an increase in genetic instability, predisposing malignant transformation. These two possibilities along with their respective risk factors and ways of risk avoidance will be discussed in this section. The recommendations formulated on the basis of the analysis are presented in Table 3.

### 4.1. De-Immortalization Efficiency: Ways to Increase and Control

Does each of the millions of cells undergo de-immortalization during immortalization reversal? The possibility of 100% efficiency of the de-immortalization process is doubtful. In the case of state-dependent de-immortalization, e.g., siRNA gene silencing (Group 1, Group 3), the reliability of shutting down the immortogene still present in the cell raises concerns. Excision of the immortalizing genetic element by site-specific recombination (the second group) mediated by transduction with Ad-Cre or Ad-FLP vectors and transduction itself are stochastic processes.

Indeed, it was demonstrated that the Tet-Off/Tet-On approach, for example, is flawed due to so-called “leakiness” caused by low-level transgene expression, even when the system is off [128,129]. A novel low-background inducible expression system modification with new TRE-promoters is currently being developed [128,129,130] because, in present transgenes, the expression under nonpermissive conditions is caused by the high basal activity of the Tet response element [130]. The recombination efficiency ranges from 63% to 86% [54], so a complete removal of immortalizing genes cannot be guaranteed. At the same time, the de-immortalization strategies in their original version do not include an efficiency control tool.

Thus, the problem is that the efficiency of de-immortalization is lower than 100% and there is no tool for efficiency control. The solution to this problem is as follows: first, the choice of the most potentially reliable de-immortalization strategy; second, increasing the efficiency of de-immortalization; third, the development of an add-on-tool for selection of cells that have been successfully de-immortalized.

#### 4.1.1. Choosing a De-Immortalization Approach

If one compares the different strategies of immortalization, gene excision appears to be a safer approach than using condition-dependent genes or siRNA-silencing. An effectively removed immortalizing gene is no longer present in a cell, which seems to be more reliable than gene “switching off” or silencing. Moreover, immortalization via site-specific gene excision makes possible a simple and trusted way to select successfully immortalized cells, as will be shown below. Thus, in our opinion, it is preferable to use the removal of the immortogene through site-specific (Cre-Lox or FLT-FRT) recombination as a method of de-immortalization.

#### 4.1.2. Tamoxifen-Mediated Self-Recombination: De-immortalization Efficiency Increase

As mentioned above, recombination efficiency depends on both proper recombination efficiency and the efficiency of Ad-Cre/Ad-FRT transduction (or transfection with Cre-/FRT-plasmid). The efficiency of the process can be significantly increased if each of the cells undergoing immortalization obviously carries the recombinase gene, the expression of which can be activated at the required stage of the process. Relevant developments are presented in a number of studies as the so-called tamoxifen-mediated self-recombination strategy. Using retroviral-mediated [131,132,133] or plasmid-mediated [43] gene transfer, a tamoxifen-inducible form of Cre (Cre-ERT2 [131,132] or ERT2-Cre-ERT2 [43,133]) is stably integrated into conditionally immortalized cell lines. Positive selection of modified cells results in stable, reversibly immortalized Cre-ERT2-(or ERT2-Cre-ERT2-)-carrying cell lines. In such a cell line, every cell carrying an immortalizing gene also already carries the Cre-recombinase gene. Combining the traditional strategy of Cre/LoxP-based reversible immortalization with a tamoxifen-regulated Cre-recombination system allows tamoxifen-dependent gene removal [43,131,132,133]. When the desired number of cells is attained, they are treated with 4-hydroxytamoxifen. In this case, Cre-ERT2 (or ERT2-Cre-ERT2) translocates from the cytoplasm to the nucleus and causes the excision of the immortalizing gene, which leads to immortalization reversion [131,132,133].

This approach can also be applied to FLT-FRT-mediated immortogene excision; however, this modification has not been implemented yet.

#### 4.1.3. Gene Excision Control

To avoid transplantation of non-de-immortalized cells, a reliable mechanism is needed to separate cells in which de-immortalization has been successful. In the case of de-immortalization via gene excision, necessary verification can easily be done. Such a methodical add-on was developed for Cre-mediated immortogene removal but can also be potentially associated with FRT-recombination.

Two approaches allow the selection of reversed cells. The first one utilizes suicide genes [8,56,107]. Suicide gene selection is based on the introduction of a viral or bacterial gene into a cell that allows the conversion of a non-toxic compound into a lethal drug [90]. There are several suicide gene systems but, in our case, the herpes simplex virus thymidine kinase HSV-tk/ganciclovir (GCV) system is used [8,107]. Herpes simplex virus thymidine kinase has a high affinity for GCV and converts it to GCV-monophosphate, which in turn is metabolized by cellular kinases to a toxic GCV-triphosphate, which causes cell death by polymerase inhibition [111]. A pair of LoxP sites flank both the immortalizing gene and the *HSV-tk* gene. Thus, after Cre-Lox recombination, the reverted cells are *HSV-tk*-negative [107], and *HSV-tk* mediates negative selection of reversed immortalization: *HSV-tk*-possitive cells will selectively die when ganciclovir is added, and *HSV-tk*-negative reversed cells will survive [8,107]. A common tool designed for this approach is the SSR#69 retroviral vector (containing the *SV40T* immortogene flanked by LoxP sites together with the *HSV-tk* suicide gene and including hygromycin-resistance gene) [43,107,110,114,115].

When the second approach is used, the immortogene is flanked by LoxP sites together with the EGFP gene. Successful recombination removes both the *EGFP* gene and the immortogene. A population of successfully reversed cells is obtained by EGFP-negative fluorescence-activated cell sorting [43,108]. A tool designed for this approach is the SSR#197 retroviral vector, where *EGFP* is necessary for both positive selection of transduced cells (firstly) and negative selection of non-de-immortilized cells (thereafter) [43,108]. A common tool designed for this approach is SSR#197 (*hTERT* flanked by LoxP sites together with *EGFP* gene) [43,108,110].

If choosing between these two approaches, preference, in our opinion, should be given to co-expression of the immortogene with *EGFP* because there is data that HSV-tk-mediated control can fail. It has been shown that Ad-mediated Cre expression can inhibit the expression of HSV-tk, giving Ad-Cre-infected cells the ability to survive even in the presence of high-dose ganciclovir [8]. Therefore, the *EGFP*-mediated approach seems to be more reliable.

### 4.2. Genetic Instability in Reversibly Immortalized Cells: Reasons and Potential Solutions

In the context of cellular biosafety, the reversibility of immortalization should be considered as a return to the original pre-immortalized state without unwanted side effects, e.g., not simply “switching off” the immortalized state of a cell. Unfortunately, the immortalization–de-immortalization process and prolonged existence in an immortalized state appear to be associated with genetic instability.

#### 4.2.1. Process-Associated Genetic Instability

Genetic instability associated with the immortalization–de-mmortalization process
First, genetic instability can be caused by the immortalization–de-immortalization process. Common approaches include using retroviral or transposon vectors to achieve integration of the immortalization gene [54]. Retroviral vectors (including lentiviral) randomly integrate into multiple sites of the host genome [134]. Transposons, according to some data, demonstrate higher site-selectivity of genomic integration [135], but specific integration sites have not been characterized [54,136]. A retroviral or transposon vector insertion into a functional gene will most likely cause a mutation and dysfunction of the gene [54,134,135]. Moreover, the Cre-Lox-system, commonly used for immortalizing and gene excision, leaves 34 bp “footprint” mutations [135,137] upon releasing the insert.

Stable, reversibly immortalized cell lines carrying Cre-ERT2 (or ERT2-Cre-ERT2) have been described above as a promising model. This strategy reduces the expense involved with making significant quantities of cells so that the potential can be widely used. However, a possible drawback of generating a cell line in which a gene encoding a Cre-recombinase is stably integrated into the genome is that a basal level of Cre expression may cause DNA breaks [138].
Genetic instability associated with immortalizing genes effects

As described above, *SV40T* [59] or *HPV16 E6/E7* [58] inserted alone (without *hTERT*), allow cells to bypass replicative senescence, but telomeres continue to shorten [7]. Telomeres play a crucial role in protecting the integrity of eukaryotic chromosomes and maintaining the genomic stability of human cells [58]. Critically short telomeres can lead to the formation of end-fusions and chromosome breakage-fusion–bridge cycles and generate additional genetic instability [7]. The validity of this reasoning can be indirectly confirmed by the fact that *SV40T*-immortalized cells demonstrated a higher number of chromosomal aberrations than *hTERT*-immortalized cells [139].

Nonetheless, *hTERT*-immortalization is associated with its own specific side effects. Telomeres carry an array of nuclear matrix attachment sites [140] and sites of binding to nuclear envelope proteins (such as lamin A, SUN2, AKTIP) [141], which are involved in structural organization and anchoring of the genome. Therefore, it is not surprising that telomere over-elongation (up to six-fold increase in the length of the telomeric region) through *hTERT*-immortalization leads to structural reorganization of the genome: chromosome regions normally located in the nuclear periphery become malposed and translocate into the nuclear interior [140]. Genome structural disorganization may cause replication and transcription disruption and lead to genome instability [142].

A significant proportion of immortalizing agents also cause the inactivation of p53 [67,68,69], which is a key player in the response to a wide range of mutagenesis events. Mutagenesis is a stochastic process that occurs when cells are constantly exposed to endogenous and exogenous factors that threaten their DNA integrity [143]. Genome stability is maintained by coordinated mechanisms known as the DNA damage response (DDR). DDRs include cell cycle arrest (which is necessary to provide time for the repair machineries to restore genome integrity), activation of the repair factor expression, or, finally, activation of the apoptotic program if DNA damage cannot be repaired [143,144]. One of the key factors of the DDR system that plays a central role in initiating the above processes is p53 [144], which is inactive in *SV40T*- or *E6/E7*-immortalized cells [67,68,69]. This obviously leads to an imbalance between damage and repair processes and mutation accumulation in these cells.

Indeed, *SV40T*-immortalized cells (as opposed to *hTERT*-immortalized ones) demonstrate a dramatically diminished DNA damage response: induced DNA damage does not lead to an increase in p53 levels and does not cause cell cycle arrest (whereas *hTERT*-immortalized cells are kept in the G1 phase of the cell cycle) [139]. In this context, *SV40T*-immortalized cells also demonstrate a large number of chromosomal aberrations [139,145].

Mitotic errors (loss of chromosomes or their parts, polyploidy, etc.) [146,147] contribute to mutagenesis and also appear stochastically. Their frequency also increases in the case of active and long-term cell proliferation of immortalized cells.

Under ordinary conditions, mitotic errors cause p53 pathway activation followed by cell cycle arrest or cell death [148,149]. However, when p53 is inactivated, cells continue to pass through the cell cycle. Mitotic errors can lead to the appearance of tumorigenic genotypes [146]. Indeed, it has been shown that cells immortalized with *hTERT* + *SV40T*, *SV40T*- or *HPV16 E6/E7*-immortolized cells (as opposed to cells immortalized with hTERT alone) demonstrated abnormal karyotypes, including complex chromosome loss [106], hypodiploid or aneuploid karyotypes [139], and features such as the absence of contact inhibition [106].

These examples suggest that *hTERT*-immortalization is more advantageous in terms of maintaining genetic stability because the process does not involve p53 inactivation. Despite this, in some cases, *hTERT*-immortalization can be associated with mutations in the p16INK4a/pRb pathways, both alleles of the *p53* tumor suppressor gene [150], and with uncontrolled activation of the *c-Myc* oncogene [151].
Genetic instability associated with culture conditions

Genetic instability in cultured immortalized cells can also be caused by currently used culture conditions, which aim to imitate a physiological environment but do not match it perfectly. Cells (which are acutely sensitive to their niche, comprised of both chemical and physical factors affecting cell behavior [152]) are cultured in vitro with a lack of extracellular matrix support and the presence of other cell types, in an abnormal mechanic environment and growth factor gradients [59,153]. Moreover, standard culture conditions were originally developed to optimize the growth of fibroblasts; thus, they utilize oxygen levels that are suitable for fibroblasts but are probably stressful for many other cell types, such as MSCs, which have been adapted to relatively low oxygen levels [153,154]. All of this leads to an increase in reactive oxygen species (ROS) during long-term cultivation. Elevated levels of ROS, in turn, are associated with higher risks of mutation and genetic instability [54,154], especially during p53 inactivation as described above.

Therefore, the process of immortalization–de-immortalization, perpetuation of the effects of genes, long-term cultivation, and culture conditions together determine the level of a cell’s predisposition to genetic instability. Furthermore, high genetic instability increases the probability of adaptive mutations, which may lead to the enrichment of a cell culture with mutant cells, as they may outcompete those without mutations during continuous active cell division [150,155].

#### 4.2.2. Reduction of Genetic Instability

The choice of immortalization–de-immortalization strategy and immortalizing agents
A key challenge of the immortalizing strategy is the need for high site-selectivity of genomic integration (to avoid host gene disruption), which is also the main trend observed. Following the use of retroviral vectors for immortalizing gene integration, the PiggyBac transposon-mediated approach (with higher site-selectivity of integration [135]) has been designed. Several studies have already used CRISPR/Cas9 system for immortalization [156], including referable immortalization [54]. CRISPR/Cas9 is the best tool for site-selective gene transfer. In this case, the immortalizing genes can be specifically targeted to so-called “safe harbors“(identified loci in human and mouse genomes) without causing significant detrimental effects to host genes [54,157].

A key challenge of de-immortalization via gene removal is the need to excise the inserts without causing footprint mutations at the excision site. To achieve seamless gene excision, the PiggyBac transposon system can be used as an immortalizing vector since it leaves no residue sequence when excised [106,137]. In addition, this previous CRISPR/Cas9-mediated gene targeting “safe harbors” at the stage of immortalization makes the presence of footprint mutations less critical, which can also be a solution.

An additional source of mutations, as mentioned above, may be basal Cre recombinase activity leading to DNA breaks [138] in “self-recombining” Cre-ERT2-expressing cell lines. There are two recommended methods to avoid this issue. The first is to use ERT2-Cre-ERT2 (not Cre-ERT2) since double ERT2 fused with Cre is shown to have the least basal activity in the absence of 4-OHT [138]. This approach has already been applied to reversible immortalization [43,133]. The second method is to put Cre-ERT2 (ERT2-Cre-ERT2) expression under the control of the Tet-On system to minimize its transcriptional activity in the absence of doxycycline [138]. However, in the context of reversible immortalization, Tet-On control seems to be an undesirable addition. This approach will reduce the level of basal expression, but a Tet-On system that has its own non-100% efficiency will also reduce the efficiency of controlled Cre-recombination required for reversing immortalization. A balance is needed between reducing Cre basal activity and maintaining high Cre-recombination efficiency, and recombination efficiency is a priority. The integration of additional massive constructs into the cell genome and their further treatment with substances (doxycycline) is also undesirable. Therefore, the use of ERT2-Cre-ERT2 seems to be sufficient.

The choice of immortogene is also an important factor. As mentioned above, *hTERT*-immortalization seems to be the safest option, and *SV40T* or *HPV16 E6/E7* immortogene usage seems to be undesirable. However, *hTERT* expression alone without *SV40T* is not enough for the immortalization of certain cell types [79]. Thus, it is necessary to find a replacement for *SV40T* to use in tandem with *hTERT*. *BMI-1* [27,35,37,145,158], *cyclin D1*, and *CDK4R24C* (mutant cyclin-dependent kinase 4, which cannot be bound by p16INK4a) [159,160,161,162] can be used for this purpose. Studies have demonstrated that *hTERT* in combination with BMI-1 as immortogenes are associated with the lowest level of chromosome aberrations among *SV40T*-, *hTERT*-, and *hTERT-BMI-1*-immortalized cells [145]. Unfortunately, in some cases, the use of *hTERT* + *BMI-1* for immortalization cannot provide a sufficient proliferation rate, and cell line developers have to reject utilizing these genes in favor of *SV40T* [27,37]. The *hTERT-cyclin D1-CDK4R24C*-immortalized cells show low incidences of chromosome abnormalities almost identical to that of wild-type cells against the background of intensive chromosome abnormalities in *SV40T*- or *E6/E7*-immortalized cells [161]. Moreover, whole-expression-pattern profiling (in immortalized corneal epithelial cells) revealed that *hTERT-cyclinD1-CDK4R24C*-immortalized cells show an expression pattern closer to that of wild-type cells, while *SV40T*-immortalized cells show an expression pattern that differs from wild-type cells [162]. This confirms that *hTERT-cyclinD1-CDK4R24C*-immortalization is gentle and does not cause significant changes to the original nature of the primary cells.
Modification of culture conditions

The level of genetic instability in cultured immortalized cells can be reduced by improving the culture conditions. Cultivation at a CO_2_ level individually selected for a particular cell type [154], the use of 3D microfluidic technologies [163], and co-cultivation with other cell types [164] (the last option is not quite suitable for immortalized cells) can be applied to cell cultures with long-term proliferation.

Another very promising approach (and a novel trend) is the use of a de-cellularized extracellular matrix (dECM). The dECM imitates the natural microenvironment of the cells [165], reproducing both its physical and mechanical properties [152], reduces ROS production [166], activates cellular antioxidant systems [167], and even demonstrates negative selection for tumorigenic cell phenotypes [59]. Moreover, dECM is also shown to participate in maintaining long-term cell proliferation. For example, dECM deposited by *SV40T*-transduced stem cells is able to support ongoing cell division of non-immortalized stem cells without losing their multipotency [153]. dECM was shown to induce sustained activation of Erk1/2 and cyclin D1, which allows the potential for some immortalizing effects [166]. Regarding the potential risks of malignant transformation and tumorigenesis associated with genetic manipulations that mediate immortalization, microenvironment-caused modulation of cell division seems to be promising [152]. This strategy, used alone, is undoubtedly not enough for primary cell immortalization, but it can be sufficient to maintain stem cells (including widely used MSCs) during serial passaging [152,153] without a decline in proliferative and differential potential, which eliminates the need for genetically induced immortalization. In the case of differentiated cells, this strategy can be used in combination with conventional genetically determined immortalization, and this may allow the use of more “soft” genetic changes, for example, increasing proliferation rate with *hTERT*-mediated immortalization.

## 5. The Need for a Tool Guaranteeing the Biosafety of Cells after Transplantation and a Promising Solution

The described modifications can significantly reduce the risk of accidental transplantation of non-de-immortalized cells and reduce the risk of unwanted mutations; however, they do not guarantee the absolute biosafety of the cell material. There is no way to control the safety of absolutely every cell. At the same time, even a single malignantly transformed or uncontrollably dividing cell can lead to the development of an oncogenic process or the formation of cell neoplasms. This gives rise to the need for an additional “line of defense” that ensures the biosafety of cells after transplantation.

A method that seems to be a good post-transplantation biosafety control option for transplanted cells was proposed by Fang and co-authors [8]. A new reversible immortalized hepatocyte cell line (HP14-19-CD) was developed in their study. HP14-19-CD cells contain two different suicide genes. The first is flanked by LoxP sites together with an immortogene, but the second is inserted separately. The first suicide gene is used for de-immortalization control and is removed together with the immortogene (as described above), while the second one remains in the cell. This gene is used for post-transplantation biosafety control. As the *HSV-tk* gene is already used in HP14-19 for immortogene insertion control, another suicide gene, the *CD* gene (codes for cytosine deaminase), is used at the second position. To implement *CD* insertion, a specific plasmid (pSEB-CD) was constructed, which also expresses a blasticidin S (BSD) selection marker and pCL Ampho mammalian expression vector.

The *CD* suicide gene is a part of the CD/5-FC system. The mechanism of the CD/5-FC system is through cytosine deaminase specifically converting the antifungal agent 5-FC to a highly toxic 5-FU, which results in the death of the target cells caused by an impairment of DNA biosynthesis [168]. Transplanted cells express the *CD* gene and can, if necessary (for example, when malignant transformation is suspected), be selectively killed by activating the CD/5-FC suicide system, which is harmless to other cells in the organism.

## 6. Conclusions

Among the considered modifications of the reversible immortalization process, several of the most promising should be noted.

CRISPR/Cas9-mediated immortogene insertion is currently presented in single studies. It has great potential because of its gentle genome editing, gene inserting into “safe harbors”, and reduced severity of “footprint” mutations. It is reasonable to reject the widely used lentivirus transduction approach in favor of CRISPR/Cas9 usage.

Tamoxifen-mediated self-recombination in stable ERT2-Cre-ERT2-expressing cell lines (or ERT2-FRT-ERT2-expressing, similarly) is another promising approach that has appeared recently. In addition to the above-described increase in the efficiency of de-immortalization (which is important in terms of both biosafety and cost-effectiveness), this strategy overcomes a formidable hurdle for the large-scale application of immortalization reversibility mediated by site-specific recombination; specifically, the need for huge amounts of viral vectors (or plasmids) [131,132] because each cell already contains recombinase.

Selection of successfully immortalized cells (by flanking an immortogene together with a fluorescent or suicide gene) is not innovative, but attention should be focused on this process one more time. In many cases, the success of de-immortalization still cannot (including resent developments) be foreseen, even though it is critically important. The possibility of convenient and reliable control of de-immortalization is the main advantage of de-immortalization via immortogene removal. This approach can be applied using any single immortogene or immortogene combination, not only for Cre-lox but also potentially for FLT recombination. This method is compatible with the two recommended approaches described above.

An important trend involves developments using dECM. This method allows not only for the provision of better physiological conditions for culturing cells but, more importantly, due to its modulating effect, it allows minimizing the genetic changes necessary for immortalization or achieving an effect without the introduction of an immortogene at all (as in the case of stem cells). However, the method is still under study. In addition, scaling up the use of the matrix in the mass production of cells seems to be technically difficult and expensive.

Finally, the method proposed by Fang and co-authors [8] to provide post-transplant safety by using suicide genes should be separately noted. The relevance and importance of this additional “line of defense” is described above. Moreover, the special significance of this approach is that it can be used as an add-on to any immortalized cell line, regardless of immortogenes or the immortalization–de-immortalization strategy, which is extremely important. Many currently existing developments (Table 1) do not meet the “safety rules” proposed in this review. For example, currently used protocols of reversible immortalization—including those based on the SV40T-immortogene (associated with an increased risk of genetic instability) and non-site-specific immortogene insertion—do not involve checking de-immortalization efficiency. They are adapted for different specific cell types (neuronal progenitors, myoblasts, cardiomyocytes, pancreatic beta cells, keratinocytes, and others) and currently are at different stages of preclinical and even clinical trials. On the one hand, cells obtained with these approaches have been successfully used without identified tumorigenic side effects in animal models and clinical trials. On the other hand, these approaches are associated with increased risks, and, as was pointed out above, even a single cell with malignant transformation or non-reversal immortalization can be enough for the development of an oncogenic process. Therefore, the challenge is to increase the biosafety of these ready-made developments without significant changes. Suicide gene insertion for post-transplantation control appears to be the best way. It can be used as an add-on to any of the existing developments, especially considering the fact that the authors have developed a specific plasmid (pSEB-CD) for suicide-gene insertion and transduced cell selection. The insertion of a suicide gene (to a safe locus) can also be carried out using the CRISPR/Cas9 approach.

We highly recommend that current or future users of the reversible immortalization method take note of the approaches provided in this review.

## Figures and Tables

**Table 1 ijms-24-07738-t001:** Immortalized cells and their potential clinical applications.

No.	Cell Type	Immortal Cell Line Name if Exists	Experimental Models or Clinical Trials	Therapeutic Effect/Potential Application	References
1	Human neural stem cells	hCTX0E03 (CTX)	Current clinical trial in patients with chronic ischemic stroke (NCT01151124, NCT03629275)	Positive therapeutic effect (upper extremity functionality enhancement) in stroke recovery	[16,17]
			Mice model, MCAo rat model	Treatment with CTX0E03 promotes angiogenesis	[18]
			Quinolinic acid-lesioned rat model of Huntington’s disease	CTX0E03 differentiate into MSNs and GABAergic neurons in the striatum; increase endogenous neurogenesis	[19]
2	Neural stem cells from human fetal spinal cord tissue	SPC-01	Rat model, balloon-induced SCI rat model	Functional recovery after spinal cord injury: differentiate into relevant ventral neuronal subtypes, reduce inflammation, reduce glial scar formation	[20,21,22]
3	Hippocampal mouse cells	MHP36	Rat model: old animals, induced brain ischemic damage	Transplanted cells repopulate hippocampal fields and areas of the cortex, differentiate into both neurons and astrocytes. Recovery of the cognitive function	[23,24,25]
4	Primary human olfactory ensheathing glia (OEG) cells	hTL4, Ts14	Rat model, induced spinal cord injury	Neuroregenerative effect: axonal regeneration of adult retinal ganglion neurons	[26,27,28]
5	Murine astrocytes (modified, GABA producing)	BASlin65	Rat model of parkinsonian tremor	Parkinson’s disease treatment. Reduce tremulous movements	[29,30,31]
6	Embryonic auditory neuroblasts	US ⁄ VOT-N33	Rat model	Replacement of auditory sensory neurons. Auditory brainstem response increase	[32,33]
7	Human fetal retinal progenitor cells	GuRt09	Neonatal hooded Lister rats, RCS dystrophic rats	Therapeutic potential in degenerative retinal diseases	[34]
8	Murine muscle progenitors (modified: transfer of human artificial chromosomes with dystrophin locus)	–	NSG mice	Duchenne muscular dystrophy gene therapy	[35]
9	Primary myoblasts	H2K 2B4	Immunodeficient dystrophin-deficient mice	Cells differentiate forming myotubes with a myogenic protein profile, regenerate host muscle	[36]
10	Primary neonatal rat cardiomyocytes	–	in vitro	Potential strategy for providing cardiomyocytes for cell therapy	[37]
11	Human pancreatic beta cells	NAKT-15	Streptozotocin-induced diabetic severe combined immunodeficient mice	Usage as a component of an implantable bioartificial pancreas. Control of blood glucose	[38,39]
12	Canine fetal pancreatic beta cells	ACT-164	Streptozotocin-induced diabetic SCID mice	Glucose dependent insulin production. Reversed chemically induced diabetes in SCID mice model	[40]
13	Murine pancreatic beta cells	–	Streptozotocin-induced diabetic mice model	Cells produce and secrete high amounts of insulin. Restore and maintain euglycemia	[41]
14	Human hepatocytes (modified, insulin-producing)	YOCK-13	Immunosuppressed totally pancreatectomized diabetic pig	Glucose dependent insulin production. Reversed chemically induced diabetes in SCID mice model	[42]
15	Human hepatocytes	16T-3	Pig model of acute liver failure	Survival prolongation	[43]
16	Human hepatocytes	NKNT-3	Rat model of acute liver failure	Usage as a component of the biohybrid artificial liver or for transplantation into the spleen. Improvement of biochemical parameters, survival increase	[44,45,46]
17	Murine fetal hepatic progenitor cells	iHPC	Recellularization the decellularized liver scaffolds in vitro	Cells efficiently recellularize decellularized liver scaffolds	[47]
18	Murine embryonic hepatic progenitors	HP14-19-CD	Nude mice, CCl4-induced acute liver injury model	Repair liver biochemical index and structure	[8]
19	Embryonic rat adrenal cells Bovine adrenal cells	RAD5.2 BADA.20	In vitro	Potential application in the treatment of chronic pain	[48]
20	Human renal proximal tubule epithelial cells	ciPTECs	In vitro, athymic nude rats	Albumin and phosphate reabsorption. Potential application as a cell component of bio-artificial kidney or in bioengineered renal tubules	[49,50,51]
21	Murine keratinocytes	iKera	Mouse skin injury model	Using cells embedded in citrate-based scaffold. Cutaneous wound reepithelialization and healing	[52]
22	Murine epidermal melanocytes	iMC	Athymic mice	Melanin production in vivo	[53]
23	Murine bone marrow stromal stem cells	imBMSCs	In vitro; nude mice, subcutaneous imBMSCs implantation		[54]
24	Human infrapatellar fat pad-derived stem cells	–	In vitro	Increase of proliferative, chondrogenic, and adipogenic abilities of IPFSCs during serial passaging	[12]
25	Murine embryonic fibroblasts (mesenchymal stem cells, MSCs)	piMEF	In vitro; nude mice, subcutaneous MSCs implantation	Osteogenic, chondrogenic, and adipogenic differentiation in vitro and in vivo, matrix mineralization in vitro	[55]
26	Primary mouse Achilles tenocytes	iMAT	In vitro	Potential application for tendon repair	[56]
27	Primary fibroblasts from adult mice	–	In vitro, Injection of iPSCs into SCID mice	Retaining reprogramming (into iPSCs) potential upon serial passaging	[57]

**Table 2 ijms-24-07738-t002:** De-immortalization approaches.

No.	De-Immortalization Approach	Strategy	Cell Types and Immortogenes	Clinical Applications Listed in:	References
Immortalization reversal mediated by changing conditions					
1	Temperature-dependent immortalization reversal	*tsA58 SV40T* immortogene (temperature-sensitive mutant allele) inactivation at body-specific temperature 37 °C (active at 33 °C cultural conditions)	*tsA58 SV40T*-transducted cells:	Table 1, No. 5, 7, 19, 20	[29,30,31,34,48,49,50,51]
			human fetal retinal progenitor cells		
			murine astrocytic transgenic cell line BAS8.1		
			embryonic rat adrenal cells		
			bovine adrenal cells		
			human renal proximal tubule epithelial cells		
			Cells from H-2K^b^-tsA58 transgenic mouse	Table 1, No. 3, 6, 9	[23,24,25,32,33,36,80,81,82]
			hippocampal cells		
			embryonic auditory neuroblasts		
			myoblasts		
			alveolar type II cells		
			Kupfer cells		
			cardiac endothelial cells		
2	Tamoxifen-dependent immortalization reversal	*c-MycER* (or *v-MycER* conjugates of *c-Myc*, *v-Myc* oncoproteins and a modified mouse estradiol receptor) inactivation at 4-OHT absence	*c-MycER*^TAM^-immortalized:	Table 1, No. 1, 2	[16,17,18,19,20,21,22,83,84,85]
			human neural stem cells		
			neural stem cells from human fetal spinal cord tissue		
			neural stem cells from human fetal spinal cord tissue		
			human late-adherent olfactory mucosa cells		
			human embryonic stem cells		
			human hepatocytes		
3	Tetracycline-dependent reversible immortalization:		*SV40T*-immortalized murine pancreatic beta cells	Table 1, No. 13	[41,86,87,88]
		Silencing of immortogene expression at tetracycline (or doxycycline) absence	*SV40T*-immortalized granulosa porcine cells		
	TeT-On	Silencing of immortogene expression by adding of tetracycline (or doxycycline)	*SV40T*-immortalized human conjunctival epithelial cells		
			*hTERT*-immortalized human bone mesenchymal stromal cells		[89]
	TeT-Off		*v-Myc*-immortalized neuronal Rat progenitor cells		
Immortalization reversal mediated by immortalizing gene excision					
4	FLT-FRP-mediated gene excision	Immortogene is flanked by FRP sites, site-specific recombination upon Ad-FLP transfection	*SV40T* (SSR#41)-immortalized murine epidermal melanocytes	Table 1, No. 22, 23, 25, 26	[53,54,55,56]
			SV40T (SSR#41)-murine bone marrow stromal stem cells		
			*SV40T* CRISPR/Cas9-mediated immortalized murine bone marrow stromal stem cells		
			*SV40T* PiggyBac transposon-mediated immortalized murine embryonic fibroblasts (mesenchymal stem cells)		
			*SV40T* PiggyBac transposon-mediated immortalized primary mouse Achilles tenocytes		
5	Cre-LoxP-mediated gene excision	Immortogene is flanked by LoxP sites, site-specific recombination upon Ad-Cre transfection	*hTERT+BMI1*-, *SV40T*- immortalized primary human olfactory ensheathing glia	Table 1, No. 4, 10, 11, 12	[26,27,28,37,40]
			*hTERT+BMI1*-, *SV40T*- immortalized primary neonatal rat cardiomyocytes		
			*SV40T*-immortalized canine fetal pancreatic beta cells		
6	Small interfering RNA-mediated gene silence	Knockdown the expression of immortogene in a sequence-specific way by mediating targeted mRNA degradation	*SV40T*-immortalized mouse keratinocytes	Table 1, No. 21	[52,90]
		siRNA-SV40T and siRNA-hTERT	*SV40T + hTERT*-immortalized human pancreatic beta cells		

**Table 3 ijms-24-07738-t003:** Reversible immortalization: recommendations for choosing a strategy that ensures maximum biosafety.

Process Stage	Key Recommendations
Immortalization	Avoid the use of *SV40T*, *E6/E7 HPV16* as immortogenes. Avoid using non-site-specific retroviral-mediated immortogene insertion. Give preference to more selectively PiggyBac- or (the best option) CRISPR/Cas9-mediated insertion. If it is enough to get the effect, use decellularized extracellular matrix (dECM) to mediate immortalization (suitable for stem cells including MSCs).
Cultivation	Use CO_2_ level selected individually for particular cell type. If it is technically possible, use dECM during culturing.
De-immortalization	Give preference to a de-immortalization strategy involving immortogene reduction followed by selection of successfully de-immortalized cells. Use PiggiBac-FLT-FRT combination for seamless immortogene removal or use CRISPR/Cas9-mediated site-spesific immortogene insertion preliminary to reduce significance of “footprint” mutations.
Pre-transplantation control	Pay particular attention to the presence of mutations in p16INK4a/pRb and p53 signaling pathways.
After transplantation	Transplanted (or used in bio-artificial organs) cells must carry a suicide gene to have an opportunity to selectively kill the cells if malignant transformation is suspected.

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
