# Peer review of "Immortalization Reversibility in the Context of Cell Therapy Biosafety"

_ijms, 2023, doi:10.3390/ijms24097738_

Round 1
Reviewer 1 Report (Previous Reviewer 1)
The text has improved considerably and in my opinion is good to go, although there are still quite a few grammatical errors which must be eliminated, much of that can be simply done using Word spell checker.
For example the last sentence reads:
We highly recommend paying attention to the listed approaches to the current or future usersof the reversible immortalization method.
What is "userof" ?
Please can the authors CAREFULLY check their text.
Author Response
Dear Reviewer, thank you very much for your comments!
We thoroughly checked once more spelling and grammer.
Reviewer 2 Report (Previous Reviewer 2)
The authors followed the guidance provided during the review process. The work has been entirely revised in the parts marked as to be corrected. There are still some repetitions in some sentences (e.g. immortalization, immortalized) and some typos to correct.
The tables have been improved and the information is easier to read.
Author Response
Dear Reviewer, thank you very much for your comments!
We thoroughly checked once more spelling and grammer.
Reviewer 3 Report (New Reviewer)

Author Response
Please see the attachment

This manuscript is a resubmission of an earlier submission. The following is a list of the peer review reports and author responses from that submission.
Round 1
Reviewer 1 Report
Sutyagina et al 2022
Not being a specialist in this particular area, I have read the review with interest. It is informative and considers am important issue of biosafety of artificially created cells proposed for transplantation in vivo
I do not have any major criticisms, only that the language of the paper needs many corrections.
The are two repetitive verbal constructions which need to be corrected.
1. Term “reimmortalization” is misleading because it means making cells immortal AGAIN. What is meant really should be called DEimmortalization, because it is a reversal of immortalization. I think this needs to be corrected everywhere or another, more accurate term used. Another suggestion is REmortalization. Check it everywhere.
2. Persistent cell lines are called “stable cell lines”, not “constant” cell lines, this is an error of translation.
Other corrections
Pg1
Abstract: Immortalization (replicative senescence genetically induced overcoming) is a perspective
Sounds clumsy. May be something like
Generically induced prevention of replicative senescence" ?
Pg 2
tissues have decreased proliferation ability because of replicative senescence
change to
tissues have LIMITED proliferation…
of their capacity to proliferate unlimitedly and generate
change to
proliferate indefinitely
Pg 3
During immortalization, cells undergo genetic remodeling which “allows overcoming”
Change to
helps to avoid
After the required cell mass has been obtain
Change to
has been obtained
necessary for “performance of their “ functions
change to
for restoration of their
pancreatic beta-cells after “immortalizing reverse” regain
change to
reversal of immortalisation
their mass production is not so difficult and expensive process.
EXPLAIN WHY?
The strategy has a wide potential application
Change to
wide range of potential applications
TABLE 1
synthesizes and adjustably
change to
releases GABA (under control of tetracycline-sensitive promoter)
GABA release in Substantia nigra pars.
WHICH PART?
Pg 6
As one can see, a wide range of cell tissue-specific types can be successfully
Change to
As can be seen, a wide range of cell types can be successfully
Pg 7
and tetracycline-dependently release GABA
change to
release GABA under control of tetracycline sensitive promoter
reversed before cells clinical application.
Change to
before clinical application of…
a number of works,
change to
number of studies
immortalization revers
change to
immporalization reversal
“bring out” associated risks,
change to
highlight
We hope that this will facilitate
Change to
We hope that this review will facilitate
2. Making cells immortalized
Change to
Immortalization of cells
Pg 8
Oncoproteins induces immortalization
Change to
Oncoproteins induce immortalization
“As the same way” human papillomaviral
Change to Similarly,
cell cycle arrest without telomere length maintaining
sense?
by preventing the maintenance of the telomere length.?
constitutive telomere “maintaining”
change to
maintenance
senescence (as known as mortality stage 1,
change to
known as mortality stage 1
making immortalization conditional “dependent” and the
change to
making immortalization conditional and the
for gene silence
change to
for gene silencing
Table 2
Title is wrong, says Table 1
Reimmortalization is not a good term (see above) May be call it REMORTALIZATION?
Approach strategy – Approach OR strategy. Not both
See for clinical application details - Clinical applications listed in:
Pg 12
at 33 C culture conditions (so-called permissive conditions),
change to
at 33 C (so-called permissive conditions),
as described before
change to
as described previously
adrenal cells 193
WHICH TYPE OF ADRENAL CELLS?
represents a “conjugate” of human c-Myc
change to
represents a fusion of human c-Myc
Pg 13
require constant treatment with tamoxifen to stop cell division
It is not tamoxifen but tet of dox which is used in this case
Pg 14
efficiency “consists of” proper recombination efficiency
change to
depends on both,
And in such cell line every
Do not start sentences with “And”…
Comparing “reimmortalisation” strategies
This term is misleading. Need something different.
Pg 15
or designing similar)
Change to
similar design
Discussing recombination-mediated gene excision in its turn it is necessary
Change to
Concerning recombination-mediated gene excision it is necessary
mediated by conditionally expressing Cre-ERT2 (or ERT2-Cre-ERT2)
change to
mediated by conditional expression of Cre-ERT2 (or ERT2-Cre-ERT2)
So even additional control is unable to grant total immortalization “reverse” and absolute biosafety of “reimmortalized” cells.
Change to
REVERSAL
And replace reimmportalized
reversibility of immortalization is understood as reimmortalization
reversibility of immortalization is a process of switching off…
gene excision leaves 34 bp [138], so-called “footprint” mutations [136] upon releasing the 356 insert.
Change to
gene excision leaves 34 bp “footprint” mutations [136] upon releasing the insert.
A constant reversibly
-STABLE
has been described above
change to
have been described above
Pg 16
number of “chromosome” aberrations than hTERT-
change to
chromosomal
In accordance with the considered examples, hTERT-immortalization…
Change to
These examples suggest that ...
Pg 17
fibroblasts so used oxygen level is suitable
change to
fibroblasts therefore used oxygen level suitable
All this leads to increasing reactive
Change to
All this leads to an increase in reactive
Change to
Therefore, the procedure of
in conditions of constant active cell divisions
change to
in conditions of constant active cell division
A key challenge of reimmortalizing, if consider gene removal
Change to
A key challenge of remortalizing(?), via gene removal
Pg 19
in the death of the target cells caused by DNA biosynthesis “impairing” [164]
change to
arrest or inhibition
Reviewer 2 Report
The manuscript entitled " Immortalization Reversibility in the Context of Cell Therapy Biosafety" is focused on an interesting topic relating to the potential new frontiers in the use of MSCs in human medicine.
There are many aspects that the authors have taken into consideration in this review and, precisely for this reason, the work should be more clearly organized.
The approach used seems to be that of a simple cataloging of possible applicable methods for cellular immortalization.
It would be better to reconsider the whole structure of the paper. The information should be organized in a more harmonious way and better describing the advantages/disadvantages of the proposed methods, focusing on the most promising and safe ones.
In the present format, the paper is not easy to read and refers to works whose result is not even briefly described. This leads to a considerable dispersion for the reader.
Authors should consider rewriting the work with a more streamlined and effective approach.
Just two examples:
Introduction
In the introduction, the authors should better outline what the purpose of the review is and how they decided to tackle the topic: which approaches they will consider and why.
Furthermore, the table inserted in the paragraph dedicated to the introduction is not clear and not even positioned correctly from a methodological point of view. It should be added subsequently.
The table is confused and contains too much information that is difficult to read.
Conclusion
The conclusions reached by the authors are very general and not aimed at what is written in the paper.
It would be more useful to better describe which are the actual most promising approaches without dispersing the information too much.